# From 2D to 3D Co-Culture Systems: A Review of Co-Culture Models to Study the Neural Cells Interaction

**DOI:** 10.3390/ijms232113116

**Published:** 2022-10-28

**Authors:** Rongrong Liu, Xiaoting Meng, Xiyao Yu, Guoqiang Wang, Zhiyong Dong, Zhengjie Zhou, Mingran Qi, Xiao Yu, Tong Ji, Fang Wang

**Affiliations:** 1Department of Histology & Embryology, College of Basic Medical Sciences, Jilin University, Changchun 130021, China; 2Department of Pathogenic Biology, College of Basic Medical Sciences, Jilin University, Changchun 130021, China

**Keywords:** neural cells interaction, co-culture system, three-dimension (3D), organoids, assembloids, microfluidic platform

## Abstract

The central nervous system (CNS) controls and regulates the functional activities of the organ systems and maintains the unity between the body and the external environment. The advent of co-culture systems has made it possible to elucidate the interactions between neural cells in vitro and to reproduce complex neural circuits. Here, we classified the co-culture system as a two-dimensional (2D) co-culture system, a cell-based three-dimensional (3D) co-culture system, a tissue slice-based 3D co-culture system, an organoid-based 3D co-culture system, and a microfluidic platform-based 3D co-culture system. We provide an overview of these different co-culture models and their applications in the study of neural cell interaction. The application of co-culture systems in virus-infected CNS disease models is also discussed here. Finally, the direction of the co-culture system in future research is prospected.

## 1. Introduction

The central nervous system (CNS) is the most complex system in the body in terms of structure and function. The basic tissue of CNS is nervous tissue and consists mainly of neurons and glial cells. Neurons and glial cells are combined in an extremely delicate and highly complex manner to form a system with three-dimensional (3D) architecture. Through complex and orderly connections between neurons or other cells, various nerve conduction pathways and neural circuits are formed, which control and regulate the functional activities of the organ systems and maintain the unity between the body and the external environment.

Thus, the understanding of the CNS cannot be limited to a single type of cell or a single organ and should be based on multiple organs at multiple levels and multiple angles. Traditional culture systems consist of a single cell type separated from the natural or complex growth environment in vivo, so the characteristics tend to be simple, and the original biological characteristics are gradually lost, resulting in physiological or metabolic properties completely different from those of natural strains or cells in vivo. As a result, it is impossible to explore the mutual relationship between multicellular systems. Moreover, information exchange and substance metabolism are needed in the microenvironment of cell survival. It has been reported that intercellular signaling plays an important role in the biological behavior of cells [1]. It is very important to study the mechanism of crosstalk between different types of cells for the study of the pathological mechanism of some diseases [2].

Cell co-culture technology can simulate the in vivo environment to a large extent, so as to better observe the interaction between cell and cell, and cell and culture environment, and explore the mechanism of drug action and possible targets by detecting the relationship between different cytokines [3]. The advent of co-culture systems has made it possible to elucidate the interactions between neural cells in vitro and to reproduce complex neural circuits. Co-culture systems can be used to study: (1) cell differentiation; (2) the function and vitality of cells; (3) cell proliferation and migration; (4) the development of CNS; (5) metabolic mechanism [4]. Although animal models are widely used in experimental studies as disease and drug screening models, in vitro co-culture models provide a more convenient and accurate system to demonstrate deeper mechanisms of interaction between cells, tissues, or organs.

This review aims to highlight the co-culture models based on two-dimensional (2D) and 3D co-culture systems related to neural cell interactions. We further discuss the recent advances in applications of organoid co-culture systems, with a particular focus on those in which a clear interaction was demonstrated through the analysis of neuromodulation applications. Their application and future directions in microfluidic platforms and disease models are also discussed.

## 2. A Brief Review of the 2D Co-Culture Model in the Research of Neural Cell–Cell Interaction

Cell co-culture models can be used to observe the interaction between cells or between cells and their surrounding microenvironment [5]. Different cell types can be plated on the same interfaces coated with an extracellular matrix (ECM) [6]. Alternatively, different cell types are placed on separated interfaces to investigate the action of certain chemical factors regulating cell behaviors [7]. These two models are named direct contact co-culture and indirect contact co-culture models, respectively [8].

For direct contact co-culture models, cells can be mixed directly or can be plated on a monolayer trophoblast. Direct contact co-culture is where two or more cells are mixed in a certain ratio and plated on the same interface under specified conditions. The most apparent advantage of this system is that it can demonstrate the interaction between glial cells and neuronal cells. For example, when neural stem cells (NSCs) were co-cultured with microglia, microglia-secreted factors enhance the dopaminergic differentiation of human NSCs [9], while co-cultured astrocytes promoted the neuronal differentiation of NSCs [10]. Furthermore, additional regulators such as immune cytokines can also be added to the co-culture system to study intercellular interactions. For example, Interleukin 33 (IL-33) was added to a mixed culture system containing primary mouse cortical neurons and glial cells to identify that IL-33 induced the release of inflammatory mediators from glial cells, thereby reducing neuronal mortality in the co-culture system [11].

A feeder cell co-culture system is a system in which cells are plated on a monolayer of certain cells (such as granulosa cells, fibroblasts, tubal epithelial cells, etc.) [12,13,14]. These feeder cells are treated with a mitotic blocker (commonly known as mitomycin) to inhibit cell division but retain the ability to secrete growth factors. The survival and proliferation of certain cells depend on growth factors secreted by feeder cells. The feeder-cell layer acts as a growth and proliferation promoter and differentiation inhibitor during the cell culture, especially in embryonic stem cells (ESCs) culture [15,16]. In addition, cellular connections are more likely to occur among multiple cell types. Based on the co-culture of two cell types, a tri-culture system consisting of neurons, astrocytes, and microglia was established. This system more realistically mimicked the neuroinflammatory response in vivo, allowing a better understanding of the influence of cellular crosstalk on neuroinflammation [3].

Cell–cell interaction or regulation is not only proceeded by the direct contact between cells but is more realized by chemical signals released in the microenvironments. In this case, a co-culture system that avoids direct contact between cells is required. Indirect contact co-culture is the cultivation of two or more different cell types so that the cells interact with each other through the chemical factors within the culture medium and without physical direct contact. The ways to achieve this may include (Figure 1): (1) Conditioned medium: cell culture supernatant containing various growth factors or stimuli secreted by cells is collected to investigate the effects of factors on cell growth or differentiation. For example, a conditioned medium collected from neurons treated with different durations of hypoxia resulted in various microglia phenotypes. (2) Feeder-cell on a coverslip: cells that secreted certain factors can also be plated on a coverslip to avoid direct contact with the cells plated in a Petri dish [17]. This method is suitable for investigating the paracrine of cells under a specific condition and can be used for quantitative analysis. (3) Transwell culture system: because of its repeatability, standardization, and simplicity, the transwell co-culture system has been widely accepted and recognized in the research on indirect intercellular interaction [18,19]. For instance, the co-culture of Schwann cells (SCs) and neurons by the transwell method have shown that beta-cellulin secreted by SCs can influence neuronal behavior and increase synapse length, thus promoting neural regeneration [20].

Although scientists have tried to reproduce the in vivo microenvironment under 2D conditions, the cells are in a 3D environment with a certain spatial structure that can influence cell behavior in vivo [21]. The data obtained from 2D co-culture conditions may have distinct results from in vivo. The growth pattern, morphology, and function of cells in 2D cultures are obviously different from those under physiological conditions in vivo: the cells show a flat growth state, abnormal division, and a possible loss of their differentiation phenotype. The abnormal cell morphology in the 2D cell culture can affect cell proliferation, differentiation, apoptosis, gene, and protein expression, and many other cellular processes [20].

## 3. Application of 3D Co-Culture Models in the Study of Neural Cells Interactions

Because it mimics an in vivo environment, the 3D co-culture system provides a reliable method for studying complex neural cell interactions, such as the synergistic and protective effects between various neural cells [22]. Three-dimensional co-culture technology can demonstrate cell activities and intercellular reactions such as differentiation and protein expression and realize real cell biology and function. Here, we classified the 3D co-culture system as a cell-based 3D co-culture system, a tissue slice-based 3D co-culture system, and an organoid-based 3D co-culture system (Figure 2).

### 3.1. Cell-Based 3D Co-Culture System

To construct a 3D architecture that simulates in vivo structure, neural cells can be encapsulated into hydrogels which have certain 3D structures after gelatinization. Both natural and synthetic hydrogels have been used for the bioengineering of 3D systems [23]. However, these biomaterials also have additional effects on cell behaviors, including cell viability, proliferation, migration, or differentiation [24]. The hydrogel biomechanical properties and the material–cell interaction may take the responsibility for this.

Matrigel is a natural hydrogel and has a positive effect on maintaining cell growth, promoting the differentiation of stem cells into neurons [25], and the extension of axons [26]. However, mouse-derived matrigel cannot be applied to clinical applications because of the uncertainty of its composition and tumor origin. In addition to Matrigel, collagen hydrogels have a similar effect on neural lineages. Yang et al. [27] reported that the combined treatment of small molecules and collagen hydrogel could induce in situ endogenous NSCs to differentiate toward neurons and restore damaged functions.

Another natural hydrogel, alginate, is also used as a potential biomaterial for constructing 3D cell culture systems since the structure of alginate is similar to hyaluronic acid, which is a primary component of brain ECM. Alginate is a by-product of iodine and mannitol extracted from kelp or the Sargasso of brown algae. The aqueous solution of alginate has high viscosity and has been used as a thickener, stabilizer, and emulsifier of food. Moxon et al. [28] mixed alginate and collagen together to create a bespoke hydrogel that mimics aspects of brain ECM. The results showed that encapsulated human pluripotent stem cells (hPSCs)-derived neurons adhere to the hydrogel matrix and formed 3D neural networks.

Recent studies have shown that natural materials, such as Matrigel and alginate can be designed, synthesized with hydrogel composites, and developed to fabricate scaffolds with pore or arrangement architectures. These structures provide the biomechanical support needed by the seeding cells on 3D scaffolds along with optimized conditions that lead to the development of functional tissue. For instance, with a defined alignment, synthesized functionalized single-walled carbon nanotube-based-alginate composite gels, Primo et al. [29] established a novel method for making electrically conductive substrates for cell therapy and other applications. In another study, adult NSCs were cultured on the carbon nanotube (CNT)–hydrogel composites, the optimized biomechanical support increased the neuron-to-astrocyte ratio and induced higher synaptic connectivity. The results indicate that hydrogel composites can be promising materials that combine high electrical conductivity with biocompatibility to promote nerve regeneration [30].

Decellularized tissue matrix (DTM) is another promising scaffold to develop personalized clinical approaches and has shown its unique and beneficial characteristics in promoting neural tissue regeneration, especially those derived from the CNS. Xu et.al [31] presented an analysis of a DTM hydrogel derived from the spinal cord (DSCM-gel). It was found that DSCM-gel retained an ECM-like nanofibrous structure and exhibited higher porosity, which potentiated NSCs/neural progenitor cells (NPCs) viability, proliferation, migration, and neuronal differentiation in the 3D culture.

In summary, hydrogel scaffolds can easily support 3D neural cell cultures. These scaffolds are porous and facilitate the transport of oxygen, nutrients, and metabolites. Thus, cells can proliferate and migrate within the scaffold network, and eventually adhere to the scaffold network (Table 1). However, the spheres obtained with this technique should be controlled in size, since a large 3D sphere can cause central necrosis due to a lack of nutrients.

### 3.2. Tissue Slice-Based 3D Co-Culture System

Organotypic brain slice culture was first developed in the 1960s. Stoppini et al. [33] introduced brain slice culture in 1991 as a new model for the research of neuroscience. To date, it has been widely used as an ex vivo model to study various aspects of neural development and regeneration, such as cell proliferation, apoptosis, embryonic cortex formation, and the migration/invasion of neural tumor cells (Table 2) [34,35,36,37,38,39]. In particular, tissue slices preserve the major advantages of in vitro systems, compensating for the lack of functionality of cell lines, and maintaining the morphological structure, tissue activity, and organ function of the tissue to some extent, thus providing a favorable microenvironment conducive to neural differentiation and neuronal circuits [40]. By co-culturing with human fetal cerebellar slices, Wang et al. [41] successfully guided iPSCs differentiation toward Purkinje neurons which possess electrophysiological functions. It has also been shown that dental pulp stem cells (DPSCs) can differentiate into neurons in the organotypic slice co-culture systems [42,43].

Another advantage of the organotypic slice is the preservation of the 3D arrangement of the vascular system in its physiological state. There is a functional interdependence between the vascular system and the CNS. Therefore, brain slices can be used as an ideal model to study the effect of the vascular system on neural differentiation and development. When ESCs were co-cultured with hippocampal slices, they differentiated into NPCs and migrated onto the vasculature of hippocampal slices. The chemokine CXCL12, produced by vascular-associated astrocytes, plays an important role in migration [44].

The combination of organotypic slices and co-culture techniques is particularly beneficial for the model of various neurological pathways and diseases. For example, under oxygen-glucose-deprived conditions, hippocampal slices can be used to model oxidative stress-caused CNS injury [45,46]. The co-culture of whole adult brain coronal slice and glioma stem cells (GSCs) can be used to model the glioblastoma tumor-host cell interactions and to study the treatment of glioblastoma multiforme [47]. In addition, brain slices are powerful tools for elucidating mechanisms of oligodendrocyte precursor cell (OPC) differentiation and myelin formation at the cellular and molecular levels. Baudouin et al. [48] transplanted OPCs into cerebellar slices to investigate myelin formation [48,49].

In addition to brain slices, spinal cord slices are also widely used in neurological research, such as neural repair and regeneration. As an important part of the CNS, the spinal cord is the pathway between the peripheral nerves and the brain; it goes up to join the medulla oblong and passes down to the peripheral nerves through spinal nerves. In one study, rat spinal cord slices were co-cultured with peripheral nerve grafts and administered with different concentrations of minocycline to observe its effect on the survival rate of motor neurons [50]. Furthermore, to analyze the interaction between vascular and neural structures, Mariya et al. [51] co-cultured mouse spinal cord slices and aortic fragments in vitro. The results showed a significant positive effect of nerve tissue on aortic sprouting. This co-culture system appears to be a useful and promising model for further investigation of the mechanisms driving the complex interactions between nerve and endothelial tissues.

Although the slice-based co-culture system provides a microenvironment highly close to the body, it still has certain limitations, such as the complex process of making and operating the tissue slice, which requires relatively fine operation and experience accumulation. In addition, like other in vitro cultures, brain slices cannot fully reproduce the physiological environment in vivo.

**Table 2 ijms-23-13116-t002:** Tissue slice-based co-culture system.

Co-Culture System	Objective	Main Results	Investigation	Refs
Purkinje progenitors and human fetal cerebellar slices.	To direct human iPSCs differentiate toward Purkinje neurons.	Fetal cerebellar slices promote differentiation and maturation of Purkinje neurons.	The degree of neuronal differentiation and electrophysiology analysis.	[41]
DPCs and adult mouse hippocampal slices.	To investigate whether human DPCs can promote neuroregeneration.	DPCs stimulated the growth of neuronal cells (especially neurons) in the edges of the hippocampal slices.	Dendrite length and cell viability.	[42]
NPCs and auditory brainstem slice.	To evaluate the potential of using DPSC as a therapeutic procedure for hearing disability patients.	Co-culture with auditory brainstem slices promotes DPSCs differentiation into neurons.	Neuronal differentiation and intracellular calcium oscillation.	[43]
ESNPs and hippocampal slices.	To explore the mechanism of ESNPs migration.	Chemokines secreted by vascular-associated astrocytes direct ESNPs migration.	Cell number and cell morphology.	[44]
MSCs and rat organotypic hippocampal slice.	To determine the neuroprotective potential of MSCs.	MSCs reduced cell death in hippocampal slices.	Cell number and function of cell secretion.	[45]
OPCs and cerebellar slice.	To investigate OPCs differentiation and myelin formation.	OPCs could efficiently differentiate into oligodendrocytes and form compact myelin in the cerebellar slice.	Myelin thickness and the area of myelinated axons.	[48]
GSCs and whole adult brain coronal slice.	To investigate distinct responses of engrafted GSCs to diverse microenvironments in the brain tissue.	Patient-derived GSCs have distinct responses to region-specific adult brain microenvironments.	Cell proliferation, differentiation, and migration.	[47]
Mouse spinal cord slices and aortic fragments.	To analyze the mechanisms of interaction between vascular and neural structures.	Nerve tissue has a significant positive effect on aortic sprouting.	Cell ratio and axon growth.	[51]

DPCs: dental pulp cells; ESNPs: embryonic stem cell-derived neural progenitors.

### 3.3. Organoid-Based 3D Co-Culture System

A new 3D model derived from PSCs, which is known as an organoid, holds great promise for modeling neural development, analyzing disease mechanisms, and developing potential therapies. Organoids, such as brain organoids and spinal cord organoids, can reproduce neural development in vitro, explore the interactions between different CNS regions, and explore the evolution of the human CNS and its unique regulatory mechanisms [52]. Thus, the organoid-based co-culture system can be developed to study the disease processes involving multiple systems or tissues, such as neuromuscular diseases (NMDs), amyotrophic lateral sclerosis (ALS) and the intricate connections between different CNS and local circuits (Table 3).

#### 3.3.1. Neuromuscular Co-Culture System

NMDs are caused by functional defects in the CNS, skeletal muscle, or the neuromuscular junction (NMJ) [53,54,55]. The NMJ is a unique, specialized chemical synapse that plays a critical role in the transmission and amplification of information from spinal motor neurons to skeletal muscle. Martins et al. [56] used hPSC-derived axial stem cells to generate human neuromuscular organoids (NMOs) that can be maintained in three dimensions for several months. The NMOs were self-organized and generated spinal cord neurons and skeletal muscle cells simultaneously. They generated contractile activity driven by functional NMJs, enabling unprecedented insight into human developmental events and the ability to analyze the role of different cell types in NMDs. The strength of this study is that these NMOs were derived directly from PSCs, giving rise to both spinal cord neurons and skeletal muscle cells, which are subjected to ectodermal or mesodermal germ layers, respectively. Future challenges will be to achieve the full maturation of NMOs and to explore how functional NMJs establish during development. For example, ALS is a progressive NMD that involves both motor neuron loss and muscle atrophy. To provide a platform to investigate the pathogenesis of ALS, Osaki et al. [57] co-cultured skeletal muscle bundles with iPSCs in the 3D matrix. NMJs can be observed in the motor neurons and skeletal muscle-formed functional connections and have synergistic effects in the 3D co-culture system [58].

Current in vitro NMJ models have been used primarily for small-scale, independent studies, allowing us to eventually explore neurodegenerative diseases of the peripheral nervous system at the molecular level and to aid in drug screening. We anticipate that future advances in co-culture systems hold promise for deepening the understanding of human NMD pathophysiology and enabling the development of practical and effective therapeutic strategies.

#### 3.3.2. Assembloids: Multi-Organism Co-Culture System

In the CNS, many nerve cells are gathered together to form an organic network or circuit. There are intricate connections between different brain regions and local circuits, which affect each other and determine the functional activity of CNS. Therefore, there is a very important feature in the completion of the above functional activities, namely, coordination and integration. These key activities have been largely inaccessible for functional studies in humans [59]. Fusing multiple organisms with different regional properties in vitro opens the opportunity for studying the interaction of specific neuronal cell types and the coordination of the organic network. Most noteworthy is that this in vitro specification of various organisms can be generated from hPSCs within personalized human micro-physiological systems. In 2017, Birney et al. [59] generated a human 3D micro physiological system that includes function-integrated glutamatergic and gamma-aminobutyric acidergic (GABA) neurons, resembling either the dorsal or ventral forebrain, to capture more elaborate developmental processes. Moreover, fusing region-specific organoids followed by live imaging enabled the analysis of human interneuron migration and integration for modeling human interneuron migration. It is worth mentioning that the neurons in fused organoids exhibited higher firing rates than the neurons in non-fused single organoids, suggesting that fused organoids confer additional neuronal properties that are not available in single organoids [60]. For instance, human medial ganglionic eminence and cortical organoids generated physiologically functional neurons and neuronal networks [61].

The most typical organoid-based co-culture system is the brain–spinal cord–skeletal muscle assembloid, which can be used to model NMDs. In 2019, Lancaster’s group published a protocol for culturing the brain organoid at the air–liquid interface and successfully used the brain organoid to control muscle contraction [62]. This system not only greatly improved the maturation and survival rate of neurons but also invaded the axons of brain organoid neurons to control the spinal cord. By controlling the spinal cord and then the muscle, the intricate connections between different CNSs and local circuits were perfectly reproduced in vitro [63,64].

**Table 3 ijms-23-13116-t003:** Organoid-based co-culture system.

Seed Cells	System Composition	Objective	Main Results	Advantages	Refs
hPSC-derived axial stem cells	NMO and NMJ.	To build NMOs and model NMDs.	A functional NMJ was generated in the constructed NMO, and a functional spinal cord network was formed.	Simultaneous differentiation of the mesoderm and ectoderm was achieved.	[56]
iPSCs	Motor neuron spheroids and 3D muscle fiber bundles.	To investigate the pathogenesis of ALS.	Formation of NMJ that can control muscle contraction.	Developed a 3D human motor unit model in a microfluidic device.	[57]
Human muscle progenitors and hPSCs	Motor neuron endplates and muscle fibers.	To model and evaluate adult human NMJ development or disease in culture.	Human muscle progenitors mixed with motor neurons self-organize to form functional NMJ connections.	Functional connectivity is confirmed with calcium imaging and electrophysiological recordings.	[58]
PSCs	3D spheroids resembling either the dorsal or ventral forebrain.	To recapitulate the saltatory migration of interneurons.	After migration, interneurons functionally integrate with glutamatergic neurons to form a microphysiological system.	The intricate connections between different CNS and local circuits were perfectly reproduced in vitro.	[59]
Human ESCs	hThOs and hCOs.	To Understand human thalamic development and model circuit organizations in the brain.	The fusion of the organoid forms a reciprocal projection.	Fused disparate regionally specified human brain organoids.	[60]
Human ESCs	Cerebral organoids, mouse spinal cord, and muscle.	To investigate whether brain organoids can produce functional neuronal output.	Cerebral organoids exhibit active neuronal networks and can innervate the mouse spinal cord.	Air-liquid interface culture of cerebral organoids leads to improved survival and maturation.	[62]

hThOs: human thalamus-like brain organoids; hCOs: human cortical-like brain organoids.

## 4. Microfluidic Platform-Based Neural-Glial Cell Co-Culture System

Elucidating intricate connections between different brain regions and local circuits requires long-term culturing and precise control. To achieve such long-term cultures, researchers have established microfluidic devices to model in vitro environments. In such devices, different types of cells are cultured in separated, interconnected chambers. Microfluidic devices enable cells to obtain nutrients and oxygen via fluid circulation and allow exposure to spatial cues or signaling gradients needed for differentiation, growth, viability, and proliferation [65,66]. Furthermore, the microfluidic platform allows the analysis of dynamic cell–cell interactions under a reproducible in vitro culture condition. In recent years, microfluidic systems have been developed for a wide range of applications in cancer research, drug screening, vascular models, and neuroscience [67].

The microfluidic system is an experimental platform that integrates the functions of driving, manipulating, monitoring, reacting, detecting, and analyzing microfluids. Generally, the system comprises different sub-systems: a fluid driving subsystem, a process monitoring, a microfluidic chip, and a detecting and analysis subsystem [68]. The microfluidic chip is the core component of the system, which usually requires the design of the microfluidic chip according to the application under study, and then the selection of other components required according to the functional requirements of the chip [69]. Microfluidic chips are characterized mainly by their effective structures (channels, reaction chambers, and certain other functional components) that contain the fluid on a micron scale in at least one latitude. Due to the micron-scale structure, the fluid displays and produces specific properties in it that are different from those of the macroscopic scale. Therefore, it has unique analytical properties. Microfluidic devices offer several advantages over traditional culture approaches, such as the spatial separation of different cell types and increased control over the cellular microenvironment (Table 4). Moreover, they are compatible with the 3D cell culture, which opens up new paths for building disease models [65].

Taylor et al. [70] used compartmentalized microfluidic chambers in an attempt to compartmentalize two distinct populations of hippocampal neurons, facilitating the visualization of synapses between them. The microfluidic compartment consisted of two main chambers connected by multiple parallel microgrooves. The study showed that neuronal axons interweaved into the microsulcus and adjacent partition chambers, indicating that synapses between two different neuronal populations can form within the microsulcus. Subsequently, Vitis et al. [71] developed a three-compartment device that can be used to control cell migration, neurite guidance between different compartments, and cell differentiation on the chip. The platform had three different perfusable compartments with distinct inlets and outlets, interconnected through a series of narrow and parallel microchannels. In this study, long-term cell culture was performed, and cells were found to differentiate into specific phenotypes in the microfluidic system.

Adriani et al. [72] developed a microfluidic device consisting of two central 3D hydrogel regions containing neurons and astrocytes mimicking neural tissues, flanked by two media channels. One of them hosted cerebral endothelial cells mimicking the blood vessel wall. Each channel communicates with the adjacent ones, offering the possibility for the cells to interact and exchange molecular cues. They demonstrated the capabilities of this 3D neurovascular model consisting of primary rat astrocytes and neurons together with human cerebral microvascular endothelial cells. In other research, Zahavi et al. [73] developed a microfluidic system to allow axonal outgrowth and the innervation of muscle in the distal compartment. This study provides a more in-depth analysis of the bidirectional molecular communication between motor neurons and muscles.

Co-culture systems based on microfluidic systems are also ideal in vitro systems as models of CNS diseases. Kunze et al. [74] developed a microfluidic system to culture healthy and diseased neurons in two separate cellular compartments. In this system, it formed neuroprotrusive network connections between them. Subsequently, a new platform is provided for the in vitro study of the interaction between affected and unaffected neurons in Alzheimer’s disease (AD). Moreover, Park et al. [75] developed a microfluidic culture system of neuronal-astrocyte-microglia that can achieve microglia recruitment, the secretion of pro-inflammatory cytokines/chemokines, and neuron/astrocyte loss, overcoming the limitation of previous neuronal models of AD that do not include neuroinflammatory changes mediated by microglia. Not only cell culture but the microfluidic system can also be applied to tissue slice culture as well. One study used a microfluidic system to culture hippocampal slices in compartments interconnected by microchannels. The slices extend axons through microchannels to form functional connections with each other, providing new ideas for studying the pathophysiological mechanisms of epilepsy and drug screening. In addition, the platform can also be used to study the pathways between other brain regions, including the limbic system pathways between the prefrontal cortex, hippocampus, amygdala, and hypothalamus, leading to a deeper insight into the brain’s information processing circuits [76].

## 5. Using Co-Culture System to Establish Microbial Infected Neural Disease Model

Certain microorganisms such as viruses can cross the blood–brain barrier and enter the CNS. Thus, the organoids can be co-cultured with microbials to establish a microbial-infected neural disease model. Here, we conclude the virus infected brain organoids in the co-culture system.

### 5.1. Using Brain Organoids Co-Culture System to Study Zika Virus-Impaired CNS

Zika virus (ZIKV) is a type of mosquito-borne flavivirus, and the infection in adults is usually mild, while vertical transmission from mother to infant causes microcephaly in newborns [77,78]. Many studies have modeled ZIKV infection by using 2D NPCs single cell culture, 3D neurospheres culture, or forebrain slice culture. However, all these models are far from disclosing the mechanism of ZIKV-induced microcephaly. In this case, the brain organoid, which can mimic neural development in vivo to some extent, is an ideal model.

Using a ZIKV-infected forebrain organoid, Haddow et al. [79] observed a significant reduction in the size of the infected forebrain organoids compared to the controls. This study suggests that upon entry into the fetal brain, ZIKV targets the proliferation of NPCs and causes small, head-like defects in cortical development. Furthermore, Garcez et al. [80] investigate the effects of ZIKV infection on neural differentiation and neurogenesis on multiple levels, including iPSC-derived NSCs, NSC-formed neurospheres, and human iPSC-derived brain organoids. The results provide more evidence to demonstrate that ZIKV inhibits neural development by targeting NPC populations and induces cell death, thereby impairing neurosphere formation. In conclusion, these studies have contributed to uncovering the effects of ZIKV on human brain development and have solidified the link between the ZIKV infection of NPCs and microcephaly in newborns. In addition to the disclosure of the mechanism, ZIKV-infected brain organoids could also be used to screen drugs for the treatment of ZIKV infection [81].

### 5.2. Virus-Brain Organoids Co-Culture System Provides Initial Insights into the Potential Neurotoxic Effects of SARS-CoV-2

The Coronavirus disease 2019 (COVID-19), caused by the severe acute respiratory syndrome coronavirus 2 (SARS-CoV-2) virus, causes potentially fatal respiratory symptoms. However, damage and dysfunction have also been found in other organs, including the kidneys, heart, liver, and brain [82]. There is growing clinical evidence of neurological symptoms, including cerebrovascular damage, altered mental status, encephalopathy, hypotactic, hyposmia, and neuropsychiatric disorders [83]. In addition, autopsy results have also indicated the presence of the virus in the brains of some patients.

Because of these superior features, brain organoids were widely used to study the location of the viral infection of the CNS and its potential targets. It has been shown that the SARS-CoV-2 virus infects the cerebral choroid plexus and can disrupt the blood–brain barrier in the brain organoid [84]. Another study used human iPSC-derived monolayers of brain cells and region-specific brain organoids to identify that the infection was associated with an inflammatory response and defective cell function. The findings support that brain organoids offer a promising tool for uncovering pathophysiological clues and potential therapeutic options for neuropsychiatric complications of COVID-19 [85,86].

It is worth noting that comprehensive research at different levels, from 2D monolayer cell models to 3D neurosphere and brain organoid models, will lead to a deeper and more thorough analysis of SARS-CoV-2 infected CNSs. Basically, the neurosphere model represents early characteristics of neurogenesis, whereas the brain organoid model exhibits features of human cortical development that recapitulate the development and physiological arrangements of the human brain [87].

It can be concluded here that co-culture systems have been widely used in the study of a variety of neurological diseases, such as ischemic stroke, Parkinson’s disease (PD), AD, and ALS [50,88,89,90,91,92]. This will likely become a trend in conducting neuroscience research in the future. With the development of co-culture and organoid modeling technologies, brain organoid-based co-culture systems will make greater contributions to the mechanistic of neurological diseases and preclinical studies.

## 6. Conclusions and Future Prospects

Co-culture systems are often designed to encapsulate the cellular interactions that occur in vivo. Investigating neural cell–cell interactions and neural circuits through a co-culture system requires complex culture conditions that meet the requirements of all involved cell types. To date, the cell co-culture system has been developed from 2D to 3D assembloids, and advances in microfluidic devices bring us closer to in vivo conditions that provide a more physiological environment to the cells. Without a doubt, the novel 3D cell culture model is an attractive method to overcome the limitations of traditional monolayer culture. In particular, 3D assembloids hold great potential in modeling various disease conditions and drug screening to study physiological and pathological cell–cell interactions. Most importantly, scientists can use patient-derived cells to generate these assembloids, generating personalized disease models to carry out individualized treatment. However, 3D assembloids still miss essential components like vasculature or immune cells. As a result, it is difficult to avoid central necrosis if we combine a number of organoids into a large single structure.

Another great challenge is the standardization of co-culture models. The combination of multidisciplinary current techniques such as culturing, biosensors, and microfluidics, could lead toward the goal of developing more complex, reproducible, nature-like in vitro tissue models. There is no doubt that multi-organ co-culture systems have been achieved in the development of various practical and theoretical research on neurodevelopment, neuromodulatory mechanisms, and neurological diseases, and have opened up new possibilities and directions.

To conclude, co-culture systems, especially the organoid-based co-culture system, are powerful tools and sets of techniques for controlling and analyzing cell interactions. However, the organoid model is the latest technology in human tissue experimental research. Compared with the traditional model, it is still in the exploratory stage. Its stability, reproducibility, scalability, and how to accurately control the microenvironmental conditions have become the problems to be overcome in the development of organoid-based co-culture technology. Most of these efforts are still proofs of principle rather than fully developed and broadly applicable alternatives to existing models, presenting inherent advantages and limitations. To create relevant co-culture systems for studies of cell interactions, the integration of organoid models in combination with standardized microdevices is desirable.

## Figures and Tables

**Figure 1 ijms-23-13116-f001:**
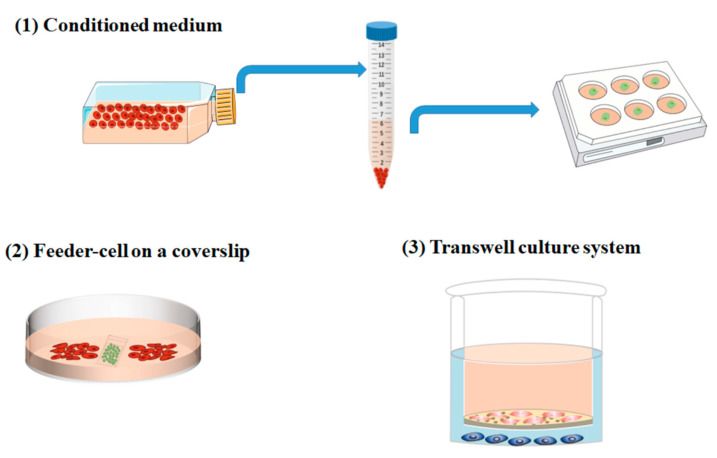
The way to achieve indirect contact co-culture: indirect contact co-culture is the cultivation of two or more different cell types so that the cells interact with each other through the chemical factors within the culture medium and without physical direct contact. The ways to achieve this may include (**1**) Conditioned medium: cell culture supernatant which contains various growth factors or stimuli secreted by cells is collected to investigate the effects of factors on cell growth or differentiation. (**2**) Feeder-cell on a coverslip: cells that secreted certain factors can also be plated on a coverslip to avoid direct contact with the cells plated in a Petri dish. (**3**) Transwell culture system: cell co-culture system in a transwell chamber.

**Figure 2 ijms-23-13116-f002:**
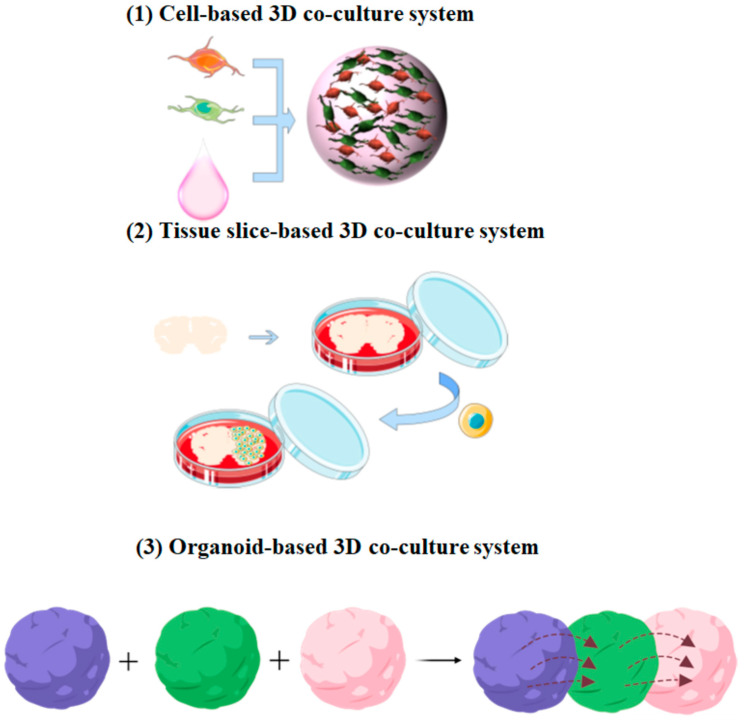
Application of 3D co-culture models in the study of neural cell interactions: (**1**) Different types of neural cells can be encapsulated into hydrogels that have a certain 3D structure after gelatinization. (**2**) Cells can be co-cultured with an organotypic slice which preserves the major advantages of in vitro system. (**3**) The organoid-based co-culture system can be developed to study the disease processes involving multiple systems or tissues.

**Table 1 ijms-23-13116-t001:** Cell-based 3D co-culture system.

Co-Cultured Cell Type	3D Construction	Effect on Cell-Interaction	Investigations	Refs
Human adult MSCs	Matrigel	Matrigel promotes cell growth and differentiation.	Cell survival rate.	[25]
SGNs	Matrigel	Matrigel promotes the survival of purified SGNs in vitro and maintained their morphological structure and function.	Neurite length.	[26]
NSCs	collagen hydrogel	Collagen hydrogel increases neuronal differentiation of NSCs and induces their migration.	Number of different types of cells, electrophysiological evaluation, and behavioral assessments.	[27]
iPSCs	Blended alginate/collagen hydrogels	The hydrogel matrix promoted neuronal differentiation and maturation.	Cell morphology and synaptophysin density.	[28]
NSCs and bone MSCs	gelatin methacryloyl	The gelatin methacryloyl promoted the generation of neurons and oligodendrocytes.	The percentage of live cells.	[32]
NSC	CNT	CNT could facilitate neuronal differentiation while maintaining neuronal homeostasis.	Cell viability and calcium imaging.	[30]
NSCs/NPCs	DTM	DSCM-gel promotes NSCs/NPCs proliferation, migration, neuron-like differentiation, and synapse formation.	Number of different types of cells and number of synapses in different periods.	[31]

MSCs: mesenchymal stem cells; SGNs: spiral ganglion neurons; iPSCs: induced pluripotent stem cells.

**Table 4 ijms-23-13116-t004:** Microfluidic platform.

System Composition	Features	Objectives	Advantages	Refs
Rat hippocampal neurons	Consists of two main chambers connected by multiple parallel microgrooves.	To visualize synapses.	Synapses originating from cell bodies in one compartment can be identified.	[70]
Neuroblastoma cell and primary SCs	Consists of three perfusable compartments with distinct inlets and outlets, interconnected through a series of narrow and parallel microgrooves.	To perform cell differentiation on the chip.	Up to three different cell populations can be cultured in a fluidically independent circuit.	[71]
Primary rat astrocytes and neurons and human cerebral microvascular endothelial cells.	Consists of two central 3D hydrogel regions and two media channels.	To assess the influence of the neurovascular microfluidic system on neural cell growth and functionality.	Supports the addition of other cell types present in the neurovasculature such as pericytes and microglia.	[72]
Motor neurons and muscle cells.	Consists of two main compartments or channels that are connected by parallel grooves.	To form a functional NMJ in the microfluidic chamber.	Allows an independent manipulation of neuronal or muscle cell populations.	[73]
Cortical neurons	Consists of two lateral cell culture channels, 24 junction channels, and the main channel.	To study interactions between healthy and diseased neurons in AD.	Connecting healthy and diseased neurons through local perfusion therapy.	[74]
Neuron, astrocyte, and microglia.	Consists of a central chamber and an angular chamber.	To model neurodegeneration and neuroinflammation in AD.	Microglia recruitment was achieved in the microfluidic system.	[75]

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
