# Peer review of "From 2D to 3D Co-Culture Systems: A Review of Co-Culture Models to Study the Neural Cells Interaction"

_ijms, 2022, doi:10.3390/ijms232113116_

Round 1

Reviewer 1 Report

In this manuscript, the authors reviewed different co-culture models, including both 2D and 3D, and their applications on the study of neural cell interaction, including discussion on virus-infected CNS 19 disease models. The manuscript is well organized and written with clarity. The authors have elaborated the advantages of the models they covered and also provided their perspectives on the currently challenges at the final section. However, it is important to provide more detail and discussion on the limitations of each model in the manuscript. This would greatly help the researchers to choose proper models to fit their studies with awareness of the caveats. 

Author Response

We appreciate for review’s suggestion and agree with reviewer that it is important to provide more detail and discussion on the limitations of each model in the manuscript. We have added these descriptions into the text (line 128-133, 137-138, 192-196, 245-249 and 465-469 in the manuscript):

Line 128-133:The growth pattern, morphology, and function of cells in 2D culture are obviously different from those under physiological conditions in vivo: the cells show flat growth state, abnormal division, and possible loss of differentiation phenotype. The abnormal cell morphology during 2D cell culture can affect cell proliferation, differentiation, apoptosis, gene and protein expression and many other cellular processes.

Line 137-138: 3D co-culture technology can demonstrate cell activities and intercellular reactions such as differentiation and protein expression and realize real cell biology and function.

Line 192-196: In summary, hydrogel scaffolds can easily support 3D neural cell cultures. These scaffolds are porous and facilitate the transport of oxygen, nutrients, and metabolites. Thus, cells can proliferate and migrate within the scaffold network, and eventually adhere to the scaffold network. However, the spheres obtained with this technique should be controlled in size, since a large 3D sphere can cause central necrosis due to lack of nutrients.

Line 245-249: Although the slice-based co-culture system provides a microenvironment highly close to the body, it still has certain limitations, such as the complex process of making and operating the tissue slice, which requires relatively fine operation and experience accumulation. In addition, like other in vitro cultures, brain slices cannot fully reproduce the physiological environment in vivo.

Line 465-469: However, the organoid model is the latest technology in human tissue experimental research. Compared with the traditional model, it is still in the exploratory stage. Its stability, reproducibility, scalability, and how to accurately control the microenvironmental conditions have become the problems to be overcome in the development of organoid-based co-culture technology.

Reviewer 2 Report

The authors present different currently developped 2D and 3D culture systems for the study of neural cells interactions. 

I think that this review is comprehensive and covers multiple aspects of the subject thoroughly. There are only some language issues that should be checked and revised.

examples:

1. Line 58: recent instead of resent

2. In some sentences passive and active voice are wrongly used as well as present and past tense are used at the same time.

ex. Line 66-67

3. Line 73: can be able: it should be can or be able etc

Author Response

We appreciate for review’s suggestion, and we have fully checked the errors and corrected them.

For example:

Line 58: We have corrected the mistake, “recent” has been instead of “resent”.

Line 66-67: We have modified this wrong expression to “Alternatively, different cell types are placed on separated interfaces to investigate the action of certain chemical factors regulating cell behaviors.”

Line 73: We have corrected the sentence, “can” has been instead of “can be able to”.

Other corrections please see the word file “ijms-1987045-with changes tracking”.
